# Roscovitine, a Cyclin-Dependent Kinase-5 Inhibitor, Decreases Phosphorylated Tau Formation and Death of Retinal Ganglion Cells of Rats after Optic Nerve Crush

**DOI:** 10.3390/ijms22158096

**Published:** 2021-07-28

**Authors:** Takahisa Hirokawa, Taeko Horie, Yurie Fukiyama, Masashi Mimura, Shinji Takai, Teruyo Kida, Hidehiro Oku

**Affiliations:** 1Department of Ophthalmology, Osaka Medical and Pharmaceutical University, Osaka 565-0871, Japan; omc122@yahoo.co.jp (T.H.); opt168@osaka-med.ac.jp (T.H.); mm080062@icloud.com (Y.F.); opt119@osaka-med.ac.jp (M.M.); teruyo.kida@ompu.ac.jp (T.K.); 2Osaka Medical College Graduate School of Medicine, Osaka 590-0906, Japan; pha010@osaka-med.ac.jp

**Keywords:** tauopathy, cyclin-dependent kinase 5 (Cdk5), roscovitine, calpain, optic nerve crush

## Abstract

Tauopathies are neurodegenerative diseases characterized by abnormal metabolism of misfolded tau proteins and are progressive. Pathological phosphorylation of tau occurs in the retinal ganglion cells (RGCs) after optic nerve injuries. Cyclin-dependent kinase-5 (Cdk5) causes hyperphosphorylation of tau. To determine the roles played by Cdk5 in retinal degeneration, roscovitine, a Cdk5 inhibitor, was injected intravitreally after optic nerve crush (ONC). The neuroprotective effect of roscovitine was determined by the number of Tuj-1-stained RGCs on day 7. The change in the levels of phosphorylated tau, calpain-1, and cleaved α-fodrin was determined by immunoblots on day 3. The expression of P35/P25, a Cdk5 activator, in the RGCs was determined by immunohistochemistry. The results showed that roscovitine reduced the level of phosphorylated tau by 3.5- to 1.6-fold. Calpain-1 (2.1-fold) and cleaved α-fodrin (1.5-fold) were increased on day 3, suggesting that the calpain signaling pathway was activated. P35/P25 was accumulated in the RGCs that were poorly stained by Tuj-1. Calpain inhibition also reduced the increase in phosphorylated tau. The number of RGCs decreased from 2191 ± 178 (sham) to 1216 ± 122 cells/mm^2^ on day 7, and roscovitine preserved the level at 1622 ± 130 cells/mm^2^. We conclude that the calpain-mediated activation of Cdk5 is associated with the pathologic phosphorylation of tau.

## 1. Introduction

Tau is a protein that stabilizes and maintains the function of microtubules in the neurons in the central nervous system (CNS), and it plays a crucial role in axonal transport. However, once axons are injured, tau is hyperphosphorylated, aggregated, and deposited in the neurons, leading to their death [1]. These changes are known to occur in cases of neuronal degeneration, and this pathological process is called tauopathy [2]. Thus, tauopathy represents pathological conditions associated with chronically progressive neurodegeneration, such as Alzheimer’s disease [3]. We have shown that tau is increased in the retinal ganglion cells (RGCs) during the process of retrograde degeneration after optic nerve crush (ONC) [4]. Because silencing the *tau* gene by a small interfering RNA (siRNA) rescued RGCs from death after ONC, tauopathy may also be associated with the changes after acute optic nerve injuries [4]. If true, then inhibition of phosphorylated tau formation may have neuroprotective effects for several kinds of optic nerve injuries in humans.

Cyclin-dependent kinase (Cdk) is a serine/threonine protein kinase that is mainly expressed in proliferating cells where it is involved in the regulation of cell cycles by controlling gene transcription and differentiation [5]. Cdk5 is predominantly located in postmitotic neurons and is associated with their development, survival, synaptic plasticity, and microtubule regulation [6,7]. Thus, Cdk5 has different characteristics from other members of the Cdk families. Hyperphosphorylation of tau is observed in the brain of Alzheimer’s patients, and some of the hyperphosphorylation is caused by Cdk5 [8]. Thus, Cdk5 is a major kinase that can cause pathological phosphorylation of tau.

Calpain is a member of the calcium-dependent protease family that is closely associated with neuronal death [9]. Calpain is known to degrade various proteins, including P35, a Cdk5 activator [10]. Because calpain is activated in the retina after optic nerve injuries, and inhibition of calpain is neuroprotective for RGCs after ONC [9], calpain may be involved in the formation of hyperphosphorylated tau [6,8].

The purpose of this study was to determine whether the calpain-induced activation of Cdk5 causes excess phosphorylation of tau that is associated with loss of the RGCs after ONC. To accomplish this, we determined the changes in the expressions of phosphorylated tau, calpain-1, and calpain-mediated degradation of α-fodrin in the retina of rats by immunoblotting after ONC. The changes in the expression of P35/P25 in the RGCs were determined by immunohistochemistry (IHC). In addition, we determined the effects of roscovitine, a Cdk5 inhibitor, and a calpain-1 inhibitor on the levels of phosphorylated tau by immunoblotting. Roscovitine was selected because it is a more potent and a more specific inhibitor of Cdk5 than other inhibitors of the Cdk families, including olomoucine [11]. The survival of RGCs was determined by IHC using Tuj-1 staining on day 7 after the ONC.

## 2. Results

### 2.1. Changes of Phosphorylated Tau after Optic Nerve Crush

We determined the levels of total and phosphorylated tau on day 3 after the ONC. Representative protein bands of total and phosphorylated tau are shown in Figure 1A, and because Ciasseu et al. showed that ocular hypertension in rats increased the levels of phosphorylated tau at ser 396 and 404 in the 50-, 55-, and 100-kDa bands in the retina [12], we measured levels of phosphorylated tau for ser 396 in the 50 kDa bands. Their levels were quantified relative to the expression of α-tubulin as an internal control (Figure 1B,C).

The results showed that both the total and phosphorylated tau levels in the 50 kDa bands were increased on day 3 after the ONC. The level of total tau was increased by 1.5 ± 0.03-fold from the sham control (*p* < 0.01, Scheffe), and roscovitine significantly reduced the increase by 0.97 ± 0.01-fold. There was no significant difference in the level of phosphorylated tau between the sham control and roscovitine-treated groups (*p* = 0.21, Sheffe, Figure 1B). The level of the phosphorylated tau was further increased by 3.5 ± 0.07-fold (*p* < 0.01, Scheffe) from the sham control level, and the intravitreal roscovitine significantly reduced the level by 1.6 ± 0.05-fold of the sham control (*p* < 0.01, Scheffe; n = 3, Figure 1C).

### 2.2. Changes of Calpain-1 and α-Fodrin after Optic Nerve Crush

Next, we determined whether calpain was activated and increased on day 3 after the ONC. Representative protein bands are shown in Figure 2A. The level of calpain-1 was increased by 3.8-fold relative to the control (*p* < 0.01, Scheffe) after the ONC. Roscovitine significantly reduced the level by 2.1-fold (*p* < 0.01, Scheffe; Figure 2B). Calpain is known to degrade α-fodrin, and there was a 1.5-fold increase in cleaved α-fodrin (150-kDa) after the ONC (*p* = 0.0096, Scheffe). Roscovitine reduced the increase in the cleaved α-fodrin by 0.8-fold (*p* = 0.002, Scheffe, Figure 2C).

Next, we determined whether the inhibition of calpain can reduce the level of phosphorylation of tau by immunoblotting. Representative protein bands are shown in Figure 3A. An increase in phosphorylated tau at ser 396 in the 50 kDa bands by 2.3-fold from the control on day 3 after the ONC was significantly reduced either by roscovitine by 1.13-fold (*p* = 0.01, Scheffe) or by the calpain inhibitor by 1.1-fold (*p* < 0.01, Scheffe, Figure 3B).

### 2.3. Expression of P35/P25 by Immunohistochemistry

Representative confocal images of flat-mounted retinas showing the time-dependent changes in the expression of P35/25, Cdk5 activators in the RGCs are presented in Figure 4. The images were taken approximately 1.0 mm from the optic disk margin. The densities of the Tuj-1 stained RGCs are high in the control retinas (Figure 4, sham) on days 1 and 3 after the ONC, while the number of Tuj-1-positive cells was clearly reduced on day 7 (Figure 4). Fine granules stained by P35/P25 antibodies were present mainly in the somas of the RGCs of the sham control. These granules were immunoreactive to P35/P25 and appeared to be coarser on days 1 and 3. On day 7 after the ONC, the fluorescein intensities of these granules appeared to be intensified in the RGCs that were poorly stained with the Tuj-1 antibody (Figure 4). The fluorescein intensities were determined by an ImageJ program and expressed in arbitrary intensity units (AIUs). The mean ± SD levels in the sham control were 12.7 ± 2.2 AIU, and they were intensified to 16.3 ± 0.9, 16.3 ± 1.7, and 19.0 ± 2.1 AIU on days 1, 3, and 7, respectively, after the ONC (Figure 4).

We also examined whether roscovitine limited the severity of immunoreactivities to P35/P25 on day 7 after the ONC (Figure 5). The results showed that the immunoreactivities to P35/P25 were intensified in the placebo group, and roscovitine reduced the intensities (Figure 5A). The fluorescein intensities increased from 13.2 ± 1.7 AIU in the sham controls to 23.0 ± 2.2 AIU in the placebo group, while the levels were significantly lower at 18.9 ± 1.1 in the roscovitine-treated group (*p* = 0,002, *t*-test). The images are shown at higher magnification in Figure 5B. The granules immunoreactive to P35/P25 were chiefly accumulated in dying RGCs that were poorly stained with Tuj-1, and roscovitine reduced the level of accumulation.

### 2.4. Effects of Roscovitine and Calpain Inhibitor on Survival of RGCs after Optic Nerve Crush

Representative confocal images of flat-mounted retinas demonstrating the effects of roscovitine and a calpain inhibitor on the survival of the RGCs on day 7 after the ONC are shown in Figure 6A. The RGCs were double-stained with Alexa 488-conjugated Tuj-1 and Alexa 555-conjugated tau. The number of Tuj-1-positive cells, most likely the RGCs [13], was reduced after the ONC, while the immunoreactivity to tau was intensified in the somas of RGCs (Figure 6A, Placebo). The fluorescein intensity of tau was 22.8 ± 2.0 AIU in the sham control, and it was significantly higher at 29.6 ± 2.0 AIU in the placebo group (*p* < 0.01, *t*-test). The fluorescein intensity of tau remained at 26.6 ± 1.9 AIU in the roscovitine and at 25.5 ± 1.2 AIU in the calpain inhibitor groups (Figure 6). Both levels were significantly lower than that of the placebo group (*p* < 0.01, *t*-tests). Although the antibody to tau binds to both total and phosphorylated tau, phosphorylated tau probably contributed more to the increased fluorescein intensity because phosphorylated tau was increased more than the total tau, as shown in immunoblotting (Figure 1). The expression of tau appeared to be increased in the somas of dying RGCs that were poorly stained with Tuj-1 (Figure 6A, placebo). These findings suggested that the increase in the expression of tau is associated with the loss of RGCs. Roscovitine appeared to reduce the degree of decrease in the number of Tuj-1-positive cells. Similarly, calpain inhibitor seemed to be protective for RGCs after the ONC.

The number of surviving RGCs on day 7 was determined using images obtained approximately 1.0 and 1.5 mm from the margin of the optic disk by fluorescence microscopy (BZ X700). The number of Tuj-1-positive cells was counted and plotted in Figure 6B. The mean ± SD number of RGCs stained by the Tuj-1 antibody was 2191 ± 178 cells/mm^2^ in the sham control (n = 4), which was decreased significantly to 1216 ± 122 cells/mm^2^ (placebo, n = 6) on day 7 after the ONC (*p* < 0.01; Scheffe). The density of RGCs was maintained at significantly higher levels of 1622 ± 130 cells/mm^2^ in the roscovitine (n = 6, *p* = 0.001) and 1671 ± 34 cells/mm^2^ in the calpain inhibitor (n = 3, *p* = 0.02, Scheffe) groups on day 7.

## 3. Discussion

The results showed that phosphorylated tau was increased in the retina after the ONC, as shown previously [4]. Roscovitine, an inhibitor of Cdk5, reduced the degree of increase in the levels of phosphorylated tau in the retina and the reduction in the density of RGCs after the ONC. There was an increase in the level of expression of calpain-1 and cleaved α-fodrin in the retina on day 3 after the ONC, suggesting that the calpain signaling pathway was activated. The expression of P35/P25 was increased chiefly in the RGCs that were poorly stained with Tuj-1. Calpain inhibition reduced the increase in phosphorylated tau and the decrease in the number of RGCs. In addition, there was a possible link between Cdk5 and the calpain signaling pathway because roscovitine reduced the increase in the levels of calpain and cleaved α-fodrin.

The concentration of roscovitine used was approximately 10 µM when the estimated volume of the vitreous cavity of rats was 20 µL [14]. This concentration was selected because the IC_50_ values for roscovitine to inhibit Cdk5/p35 are reported to be 0.16 µM. In addition, other kinases tested were less sensitive to 10 µM roscovitine [11]. A similar concentration of roscovitine is widely used as a selective Cdk5 inhibitor in the CNS [15,16], and it was reported to decrease the levels of phosphorylated tau and preserved the number of Tuj-1-positive RGCs. These findings suggested that Cdk5 participated in the pathological phosphorylation of tau after the ONC. The effects of roscovitine after neuronal insults would be more important from a therapeutic point of view. It has been shown that Cdk5 activity is already intensified at 3 h after induction of brain ischemia [17]. Thus, we injected roscovitine just after the ONC.

The activation of calpain by the influx of Ca^2+^ ions is known to occur in the retina after various types of insults, including ischemia-reperfusion [18], ONC [9,19], and excitotoxicity [20]. Calpain inhibition rescues RGCs from these insults [9,18,19]. In addition, the level of calpastatin, an endogenous inhibitor of calpain, is reduced in degenerating axons, and supplementation of calpastatin lowered axonal degeneration after optic nerve injuries [21]. Thus, the calpain-mediated signaling pathway is playing a crucial role in the death of RGCs from optic nerve injuries, retinal ischemia, and excitotoxicity. Our results suggest that calpain inhibition may decrease the phosphorylation of tau in the retina, and it may be one of the possible mechanisms of the neuroprotective effects of calpain inhibition after ONC. Hung et al. have shown similarly that calpain inhibition can depress the Cdk5 activation and hyperphosphorylation of tau after spinal cord injuries [22].

A cytoskeletal protein of fodrin is a major substrate of calpain, and the increase in the 150-kDa fragment of cleaved α-fodrin suggested that the calpain signaling pathway was activated [23]. Bax, procaspases, and PARP can also be cleaved by calpain [24], which may account for calpain’s action of causing apoptosis. A Cdk5 activator P35 is changed to P25 by calpain that binds more tightly to Cdk5, causing a stronger activation of Cdk5 [8,25]. However, we could not detect significant changes in the P35 levels in the retina and also could not detect degradation of P35 to P25 by immunoblotting (data not shown). Our inability to detect P25 on day 3 after the ONC is consistent with previous reports that showed the neuroprotective effects of Cdk5 inhibition on RGCs from optic nerve transection [26] and experimental glaucoma in rats [27]. The findings of earlier reports suggested that activation of Cdk5 occurred very quickly after the optic nerve injuries [28], and degradation may transiently occur at earlier times [27]. On the other hand, our immunohistological study showed time-dependent changes in the expression of P35/P25 granules in the RGCs, and the distribution of the P35/P25-positive granules was uneven and accumulated in the RGCs that were poorly stained by Tuj-1. These results indicated that the P35/P25 proteins, activators of Cdk5, may be involved in the death of the RGCs. In addition, ONC primarily involves the RGCs in the pathologic processes, and changes of P35/P25 may be localized in the RGCs. This may also account for the absence of significant changes of P35/P25 expressions in the whole retinal lysates by immunoblotting.

On the other hand, the inhibition of Cdk5 by roscovitine reduced the activation of the calpain signaling pathway, as shown by a decrease in the levels of calpain-1 and cleaved α-fodrin. Thus, there may be a cross-reaction between Cdk5 and the calpain signaling pathway. In this regard, it has been suggested that Cdk5 enhanced retinal injuries after activation of the NMDA receptors, which in turn increased the influx of Ca^2+^ ion and calpain activation [15]. It has also been shown that roscovitine can inhibit some calcium channels [16].

There are several limitations in this study. One limitation was that we did not determine how Cdk5 affects the formation of hyperphosphorylated tau and tau oligomers, as they are more responsible for the pathogenesis of neurodegeneration. Another limitation was that although cell cycle proteins are generally suppressed in the mature neurons, neuronal injuries may activate cell cycle pathways and phosphorylate retinoblastoma protein, leading to neuronal death [29]. It has also been shown that Cdk5 mediates excitotoxicity-induced neuronal death [15,30], and roscovitine may rescue RGCs independently of tau phosphorylation. The ONC also causes glial activation and neuroinflammation, which are associated with cell cycle activation. Roscovitine may inhibit these cell cycle-related events and exert neuroprotection [29]. On the other hand, roscovitine is widely used as a Cdk5 inhibitor in the CNS, and it can also inhibit other Cdk families, including Cdk2 [11]. In addition, glycogen synthase kinase (GSK) 3β is reportedly involved in the formation of hyperphosphorylated tau in the retina [31], and Cdk5 activation may affect the activities of GSK 3β [32]. These issues need to be investigated further.

A schematic summary is shown in Figure 7. Cdk5 inhibition by roscovitine decreased the level of phosphorylated tau in the retina and preserved the number of RGCs after the ONC. In the mechanisms, the inhibitory effects on the calpain signaling pathways may be involved. These findings indicate that therapeutic interventions that decrease phosphorylated tau levels may be helpful for optic nerve injuries.

## 4. Materials and Methods

### 4.1. Animals

Nine-week-old male Wistar rats were purchased from Japan SLC (Shizuoka, Japan) and housed in an air-conditioned room with a temperature of approximately 23 °C and 60% humidity. The room lights were set on a 12:12 light:dark cycle. All animals were handled in accordance with the ARVO Statement for the Use of Animals in Ophthalmic and Vision Research. The experimental protocol was approved by the Committee of Animal Use and Care of the Osaka Medical College (No. 28024).

### 4.2. Chemicals

Unless noted, all chemicals were purchased from Sigma-Aldrich Corp. (St. Louis, MO, USA).

### 4.3. Anesthesia and Euthanasia

All surgeries were performed under general anesthesia induced by an intraperitoneal injection of a mixture of medetomidine (0.75 mg), midazolam hydrochloride (4.0 mg), and butorphanol tartrate (5.0 mg/kg body wt). Rats were euthanized in a 13.8 L cage with wood-shaving bedding by exposure to CO_2_ at a rate of 6 L/min.

### 4.4. Optic Nerve Crush

Animals were anesthetized, and an incision was made along the midline of the skull to expose the superior surface of the left eye. The superior rectus muscle was incised to expose the left optic nerve, and the left optic nerve was crushed 2 mm behind the eye with forceps for 10 s. Care was taken not to occlude the blood vessels and cause retinal ischemia. We confirmed that our procedure did not cause retinal ischemia by ophthalmoscopic examination and by showing that the HIF-1α gene was not up-regulated by real-time PCR [33]. A sham operation was performed on the left eyes of other animals (sham controls) by exposing the optic nerve in the same way, but the optic nerve was not crushed. The right eyes were untouched in all animals.

To determine the effects of the Cdk5 inhibitor and calpain-1 inhibitor, 10 µM roscovitine and 200 µM calpain-1 inhibitor 1 (Roche, Basel, Switzerland, ref. 11086090001) were injected into the vitreous of rats just after the ONC. Intravitreal injection of 3 µL of dimethyl sulfoxide (DMSO) was performed on the placebo controls just after the ONC.

### 4.5. Immunohistochemistry

Earlier studies have shown that axotomy or optic nerve crush causes a delayed death of RGCs; the number of RGCs is unchanged for 5 days after the injury and then decreases to 50% on day 7 and to less than 10% on day 14 [34]. Thus, the number of surviving RGCs was determined on day 7 after the ONC.

Rats were euthanized as described, and the retinas were carefully removed from the eyes according to the methods described in detail by Winkler [35]. The isolated retinas were fixed in 4% PFA in PBS overnight. After washing in PBS and blocking in PBS containing 1.0% BS and 0.3% Triton X-100, the retinas were incubated with Alexa 488-conjugated mouse monoclonal neuron-specific class III β-tubulin (1:500 Tuj-1) antibody (Biolegend, San Diego, CA, USA). Tuj-1 is a marker for RGCs [36]. The retinas were placed in the same medium overnight at 4 °C and washed with PBS and cover slipped the next morning.

To determine the number of RGCs surviving, the stained flat mounts were photographed with a fluorescence microscope (BZ X700; Keyence, Osaka, Japan). Eight areas, each 0.48 × 0.48 mm, from the four quadrants in each direction of the retina at 1.0 and 1.5 mm from the margin of the optic disc were photographed. All the Tuj-1-positive cells in an area of 0.2 × 0.2 mm at the center of each image were counted using the ImageJ program (http://imagej.nih.gov/ij/; provided in the public domain by the National Institutes of Health, Bethesda, MD, USA).

The mean density of the Tuj-1-positive cells/mm^2^ was calculated, and the loss of RGCs was estimated by comparing the density in the retinas of experimental groups to that in the sham control group. The number of Tuj-1-positive cells was counted by one observer (TH) who was masked as to whether it was from an experimental or a sham animal.

In addition to the Tuj-1-positive cells, retinas were incubated with Alexa 555-conjugated rabbit monoclonal antibody to tau (1:100, Abcam, Cambridge, MA, USA) overnight at 4 °C. This antibody binds to total tau and is not specific to phosphorylated tau. In some experiments, the time-dependent changes in the expression of P35/P25 in the RGCs were determined using rabbit polyclonal P35/P25 antibody (1:100, Cell Signaling, Danvers, MA, USA) and Alexa 594-conjugated goat anti-rabbit IgG (1:500, Abcam). The retinas were photographed with a confocal laser microscope (TCS SP8, Leica, Wetzlar, Germany). The level of expression of tau and P35/P25 was determined by their fluorescein intensities and semi-quantified using the ImageJ program (n = 8–10 images in each experimental group).

### 4.6. Protein Extraction

After euthanizing the animals at the selected times, the retinas were isolated, and 3 retinas in each experimental group were pooled. They were homogenized in RIPA buffer constituted of 50 mM Tris/HCl (pH 7.6), 150 mM NaCl, 5.0 mM KCl, 0.5% sodium deoxycholate, 1.0 mM EDTA, 1.0% nonidet P40, and 1.0% protease inhibitors (Nacalai tesque, Kyoto Japan) in ice-cold water. After centrifuging at 10,000× *g* for 10 min at 4 °C, the supernatant was collected. The total protein concentration was measured by the Bio-Rad protein assay (Bio-Rad, Hercules, CA, USA).

### 4.7. Immunoblotting for Expression of Phosphorylated Tau after Optic Nerve Crush

Immunoblotting was performed to determine the changes in the levels of expression of phosphorylated tau. The phosphorylation of tau is known to occur at many sites, among which phosphorylation at ser 396 and 404 are one of the earliest events in Alzheimer’s disease, which causes tau aggregates in the brain [37]. It has been shown that phosphorylation at these sites occurs in the retinas of rats with experimental glaucoma [12], and pseudo-phosphorylation at these sites causes polymerization of tau [37]. Thus, we used rabbit monoclonal antibody to the phosphorylated tau at ser 396 (Abcam) for immunoblotting. We showed earlier that the phosphorylated tau at ser 396 increased in the retina on day 7 after the ONC [4]. Because the loss of RGCs becomes evident by day 7, the damaging effects of phosphorylation of tau would occur at earlier time points. Thus, we determined the levels of phosphorylated tau at ser 396 on day 3. In addition, because calpain is reported to play a role in the phosphorylation of tau, the levels of calpain-1 and its substrate α-fodrin were measured by immunoblotting. For this, samples were separated on a 7.5% SDS-polyacrylamide gel and transblotted onto PVDF membranes. The membranes were blocked with 5% skim milk in TBS-T (pH 7.4, 0.1% Tween 20) followed by overnight incubation with a rabbit monoclonal anti-phosphorylated tau (ser 396, Abcam) and mouse monoclonal anti-total tau (A-10, Santa Cruz, Dallas, TX, USA), rabbit polyclonal anti-calpain-1 (Abcam), and rabbit monoclonal anti-α-fodrin (Abcam) antibodies at 4 °C. α-tubulin (Merck Millipore, Darmstadt, Germany) was used as an internal control. Immunoblots were performed using pooled samples composed of 3 retinas in each group, and measurements were made in triplicate (n = 3).

The protein bands were made visible by horseradish peroxidase-conjugated to appropriate secondary antibodies (Promega, Madison, WI, USA). The signals were intensified with an ECL plus Western blotting detection system (GE Healthcare, Buckinghamshire, U.K.). The densities of the bands of proteins were quantified with a luminescent image analyzer (LAS-3000, Fujifilm, Tokyo, Japan). The levels of expression of these proteins were quantified with the embedded software (Multi-Gauge version 3.0) and standardized to the control level.

### 4.8. Statistical Analyses

The data are expressed as the means ± standard deviations (SDs). Statistical analyses were performed by one-way analysis of variance (ANOVA), and if significant changes were detected, the Scheffe post-hoc test was used. When comparison was made between 2 groups, Student’s *t*-test was used. The level of significance was set at *p* < 0.05.

## Figures and Tables

**Figure 1 ijms-22-08096-f001:**
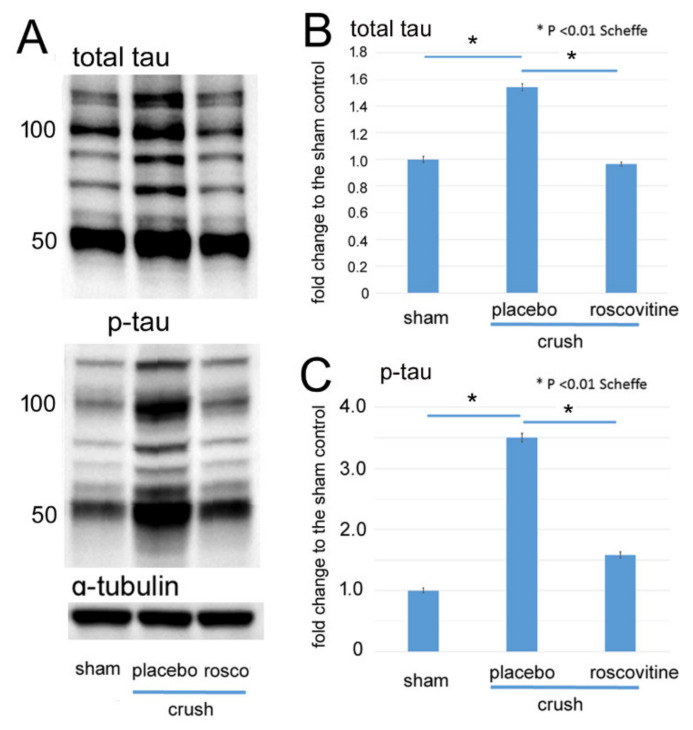
Western blots for phosphorylated tau in the retina after ONC. (**A**): Representative protein bands of total and phosphorylated (ser 396) tau in extracts from the retina on day 3 after ONC. α-tubulin was used as internal control. (**B**,**C**): Levels of total (**B**) and phosphorylated tau (**C**) in 50-kDa bands are shown as fold changes (means ± SD) relative to the sham control. Both levels significantly increase in the retina after ONC (placebo) from the sham control, while roscovitine reduced the increases. (* *p* < 0.01, Scheffe; *n* = 3 for each condition).

**Figure 2 ijms-22-08096-f002:**
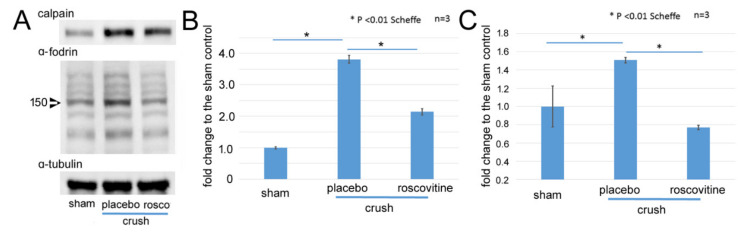
Western blots for calpain-1 and cleaved α-fodrin in the retina after ONC. (**A**): Representative protein bands of calpain-1 and its substrate α-fodrin in extracts from the retina on day 3 after ONC. α-tubulin was used as internal control. (**B**,**C**): Levels of Calpain-1 (**B**) and cleaved α-fodrin in 150-kDa bands (**C**) are shown as fold changes (means ± SD) to the sham control. Both levels significantly increased in the retina after ONC (placebo), while roscovitine reduced the degree of increase. (* *p* < 0.01, Scheffe; *n* = 3 for each condition).

**Figure 3 ijms-22-08096-f003:**
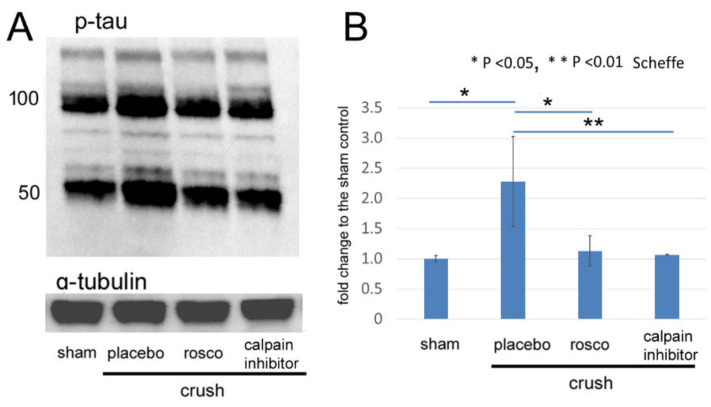
Western blots for phosphorylated tau in the retina after ONC and effects of roscovitine and calpain inhibitor on the degree of expression. (**A**): Representative protein bands of phosphorylated (ser 396) tau in extracts from the retina on day 3 after ONC. α-tubulin was used as an internal control. (**B**): Phosphorylated (ser 396) tau levels in 50-kDa bands are shown as fold changes (means ± SD) relative to the sham control. The levels significantly increased in the retina after ONC (placebo) from the sham control, while roscovitine and calpain inhibitor reduced the degree of increase. (* *p* < 0.05, ** *p* < 0.01, Scheffe; *n* = 3 for each condition).

**Figure 4 ijms-22-08096-f004:**
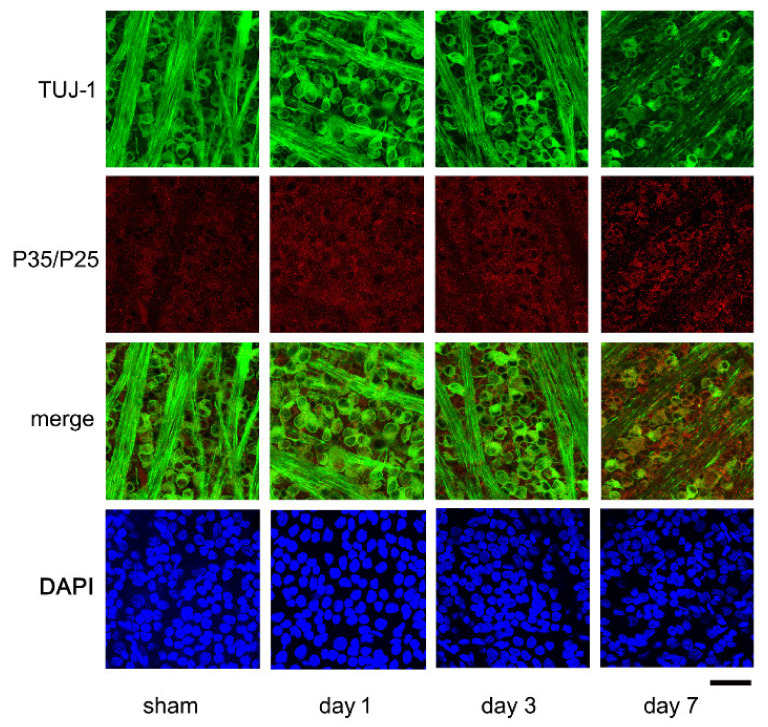
Representative confocal images of flat-mounted retinas demonstrating time-dependent changes in Tuj-1 and P35/P25 proteins. Tuj-1-stained cells (green), most likely RGCs, are densely packed in the sham control retina. Fine granules stained with P35/P25 (red) are distributed mainly in the somas of these cells. Retinal nerve fibers are also stained with Tuj-1. While the number of Tuj-1-stained cells seems to be stable on days 1 and 3, there are time-dependent changes in the expression of P35/P25, and the P35/P25-positive granules are coarse with increased fluorescein intensities. On day 7, the immunoreactivities to Tuj-1 in some cells are lost where P35/P25-positive granules are accumulated, suggesting pathological roles of P35/P25 proteins. (Bar = 100 µm).

**Figure 5 ijms-22-08096-f005:**
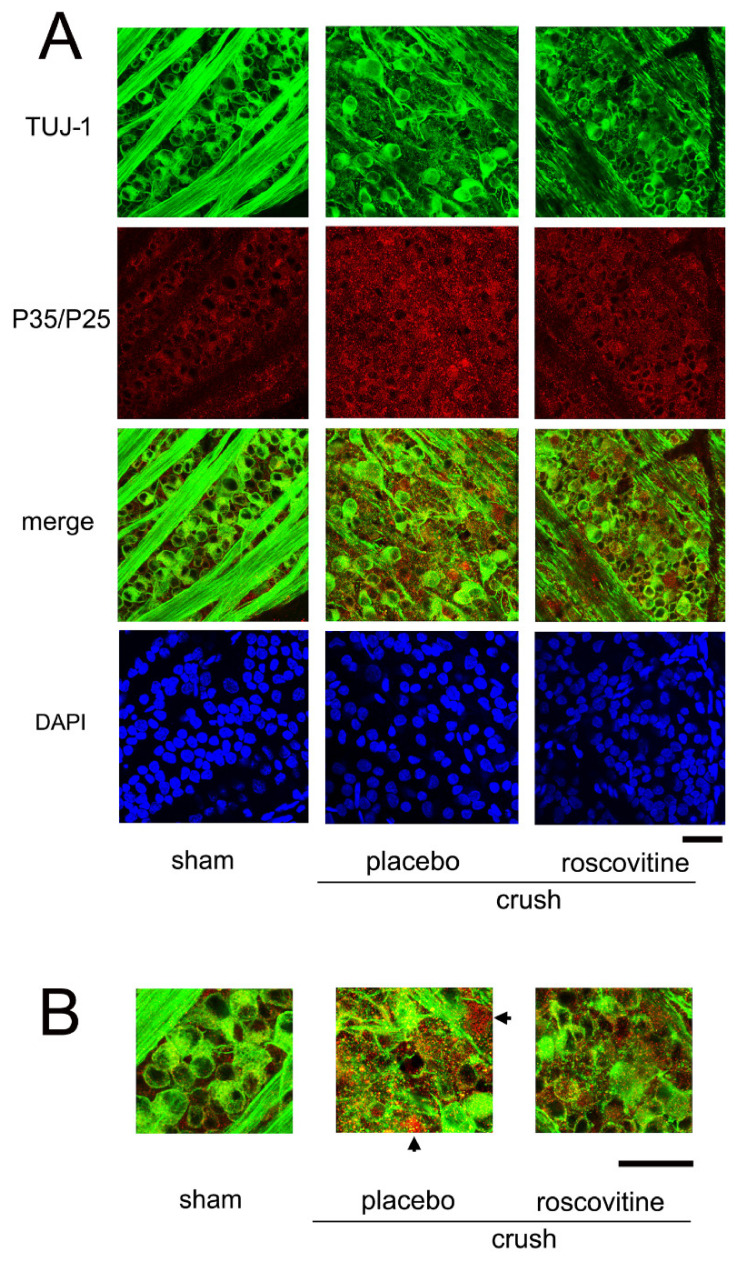
Representative confocal images of flat-mounted retinas demonstrating effects of roscovitine on the changes in the Tuj-1 and P35/P25 proteins. (**A**). Double labeling for Tuj-1 and P35/P25 in each group is shown. P35/P25 immunopositive granules are distributed unevenly with higher intensities after ONC (placebo). (**B**). Merged images are shown with higher magnification. P35/P25 immunopositive granules are accumulated in the dying RGCs where immunoreactivities to Tuj-1 are lost in the placebo group (arrows), while immunoreactivities to P35/P25 are less intense in the roscovitine-treated group. (Bar = 100 µm).

**Figure 6 ijms-22-08096-f006:**
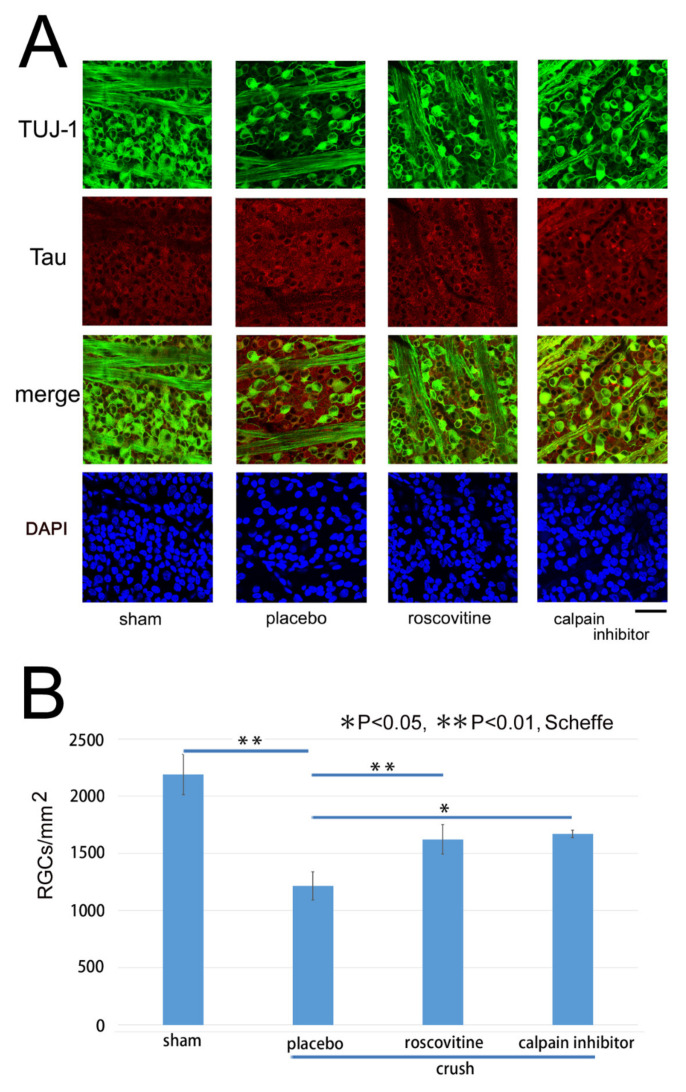
Representative confocal images of flat-mounted retinas double-stained with Tuj-1 and tau antibodies demonstrating the protective effects of roscovitine and calpain inhibitor on retinal damages. (**A**): The number of RGCs stained with Tuj-1 (green) is clearly reduced after the ONC (placebo), while roscovitine and calpain inhibitor preserve the number of RGCs. The expression of tau (red) appears to be intensified in the RGCs where immunoreactivities to Tuj-1 are poor after ONC (placebo). Roscovitine and calpain inhibitor seemed to reduce the degree of increase in tau expression after ONC. (Bar = 100 µm). (**B**): Number of Tuj-1 positive cells most likely the RGCs on day 7 after the ONC. RGCs are significantly fewer than that of the sham control on day 7 (placebo), and roscovitine and calpain inhibitor significantly reduce the degree of decrease. (* *p* < 0.05, ** *p* < 0.01, Scheffe; *n* = 3–6 in each condition).

**Figure 7 ijms-22-08096-f007:**
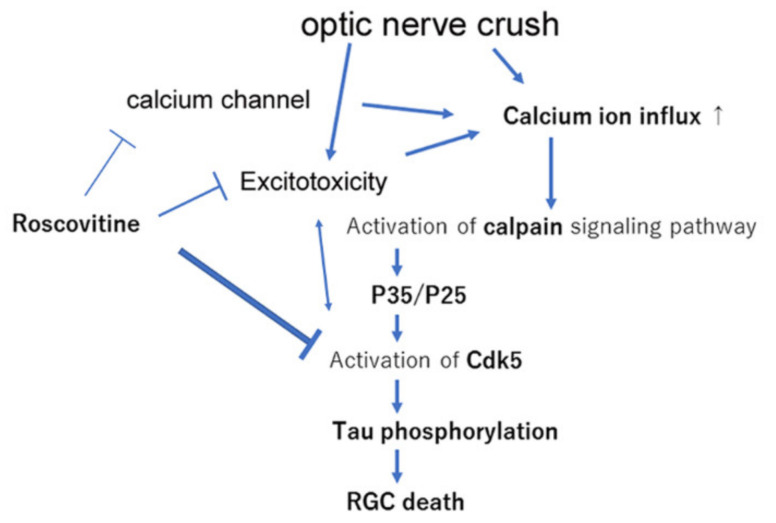
Schematic diagram illustrating the effects of roscovitine. Optic nerve crush increases calcium influx and activates the calpain signaling pathway, which then activates Cdk5 through a mechanism involving P35/P25. Activation of Cdk5 causes abnormal phosphorylation of tau, leading to the death of the RGCs. Roscovitine reduces the degree of activation of Cdk5 and decreases the abnormal phosphorylation. Cdk5 is also associated with excitotoxicity, and roscovitine may decrease calcium influx through a mechanism involving excitotoxicity. In addition, roscovitine may block some calcium channels and thereby suppress the calpain signaling pathway.

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
