# Peer review of "Roscovitine, a Cyclin-Dependent Kinase-5 Inhibitor, Decreases Phosphorylated Tau Formation and Death of Retinal Ganglion Cells of Rats after Optic Nerve Crush"

_ijms, 2021, doi:10.3390/ijms22158096_

Round 1

Reviewer 1 Report

In their study, Hirokawa and colleagues hypothesized that the calpain-induced activation of Cdk5 causes an excess of tau phosphorylation that is associated with loss of the retinal ganglion cells (RGCs) after optic nerve crush (ONC). Therefore, they assessed tau levels and phosphorylation, calpain and α-fodrin levels, P35/P25 staining, and RGCs viability in rats after ONC, with and without treatment with roscovitine, a Cdk5 inhibitor.

Key results were that:

-           phosphorylated tau was increased in the retina after ONC

-           roscovitine and calpain inhibition reduced phosphor-tau in the retina after ONC

-           levels of calpain, cleaved É‘-fodrin and P35/P25 were increased, indicating that the calpain signalling pathway was activated

The data also suggested a possible link between Cdk5 and calpain signaling pathway, as roscovitine reduced the levels of calpain and cleaved É‘-fodrin after ONC.

Major concerns:

  1. I have noticed typo / grammatical mistakes (e.g. line 69 “roscovitine significantly depressed the increase…”) throughout the text. I highly recommend the authors to have the article proof-read by a native English speaker.
  2. The abstract may be re-shaped in order to include sentence about the study background, which may help to understand the purpose of the study.
  3. Authors should emphasize in the introduction that CdK5 is one the main kinase involved in abnormal Tau phosphorylation. They should also put their study in a more general context (therapeutic intervention in which situation?)
  4. The number of replicates per group is very low (n=3).
  5. Why authors choose to use roscovitine at the first place, and not another Cdk5 inhibitor or another inhibitor of tau phosphorylation ?
  6. Did the author check if the fluorescence intensity of P35/P25 correlated with tau staining over time (from day 1 to day 7)? If yes, this might suggest a correlation between Cdk5 activation and tau phosphorylation, and emphasize the link with RGC death.
  7. Excessive Cdk5 activation was shown to induce neuronal apoptosis. Can the loss of RGCs be due to this over-activation, independently of tau phosphorylation?
  8. Which tau antibody was used for the staining of RGCs (Figure 6A)? Antibody for total of phospho-tau? At which day were the image taken? Does it correspond to the quantification in Figure 6B (day 7)?
  9. Calpain activation is up-stream to Cdk5 activation. How to explain that roscovitine decreases calpain levels and activity (decreased cleaved É‘-fodrin)?
  10. The statistical analysis is missing in the method part.
  11. In order to increase the general understanding of the study, I would advise the authors to include a summary scheme showing the pathway (calpain / P35/P25 / Cdk5) they described in their study and how roscovitine is acting on this pathway.

Author Response

Responses to the Reviewer 1.

We have revised the manuscript according to your comments and suggestions. We thank you for your constructive comments. Our responses are detailed below.

  1. I have noticed typo / grammatical mistakes (e.g. line 69 “roscovitine significantly depressed the increase…”) throughout the text. I highly recommend the authors to have the article proof-read by a native English speaker.

Answer: The phrase of “depressed the increase” was changed to “reduced the increase” throughout the text. The manuscript was thoroughly edited by a native English speaking Professor Emeritus Duco Hamasaki of the Bascom Palmer Eye Institute, University of Miami School of Medicine.

  1. The abstract may be re-shaped in order to include sentence about the study background, which may help to understand the purpose of the study.

Answer: The abstract was entirely revised, and the background of the study has been added.

  1. Authors should emphasize in the introduction that CdK5 is one the main kinase involved in abnormal Tau phosphorylation. They should also put their study in a more general context (therapeutic intervention in which situation?)

Answer: In the Introduction, we state that Cdk5 played a critical role in the abnormal phosphorylation of tau in Alzheimer’s disease (line 45-48). In addition, the last sentence in the Abstract was changed to “calpain-mediated activation of Cdk5 is associated with pathologic phosphorylation of tau”.

  1. The number of replicates per group is very low (n=3).

Answer: To minimize the sacrifice of animals is important. In the immunoblotting, we pooled 3 retinas from 3 animals and the assays were done in triplicate. We also performed immunoblotting on day 7 and obtained similar results but these data are not shown. Thus, we believe the results are solid.

  1. Why authors choose to use roscovitine at the first place, and not another Cdk5 inhibitor or another inhibitor of tau phosphorylation ?

Answer: Roscovitine is a more potent and selective inhibitor of Cdk5 than other inhibitors in the Cdk families including olomoucine. This information was added to the Introduction (line 61-62) and also in the Discussion (line 206-210).

  1. Did the author check if the fluorescence intensity of P35/P25 correlated with tau staining over time (from day 1 to day 7)? If yes, this might suggest a correlation between Cdk5 activation and tau phosphorylation, and emphasize the link with RGC death.

Answer: We measured the fluorescein intensities of P35/P25 and tau (line 344-346) and the results have been added in the Results section (line 127-129, 143-145, 163-167).

  1. Excessive Cdk5 activation was shown to induce neuronal apoptosis. Can the loss of RGCs be due to this over-activation, independently of tau phosphorylation?

Answer: We agree that this possibility cannot be ruled out, and we have stated that activation of Cdk 5 independent of tau phosphorylation could cause the death of RGCs as a limitation of this study (line 259-262).

  1. Which tau antibody was used for the staining of RGCs (Figure 6A)? Antibody for total of phospho-tau? At which day were the image taken? Does it correspond to the quantification in Figure 6B (day 7)?

Answer: The antibody to tau used in the IHC binds to total tau and is not specific to phosphorylated tau (line 339). However, the level of phosphorylated tau increased more than total tau as shown in the Figure 1. Phosphorylated tau may contribute more to the increase of the fluorescein intensities in Figure 6 (line 167-170). The results shown in Figure 6 are obtained from retinas on day 7 after the ONC and the results in Figures 6A and 6B are corresponded.

  1. Calpain activation is up-stream to Cdk5 activation. How to explain that roscovitine decreases calpain levels and activity (decreased cleaved É‘-fodrin)?

Answer: Roscovitine reduced the activation of calpain signaling pathway although the exact mechanism(s) has not been determined. As described in the Discussion section, roscovitine might reduce the degree of excitotoxicity and Ca2+ influx. In addition, roscovitine might control some calcium channels and thereby inhibit calpain activities (line 251-254).

  1. The statistical analysis is missing in the method part.

Answer: We now include descriptions of the Statistical analyses in the Methods section (line 386-390).

  1. In order to increase the general understanding of the study, I would advise the authors to include a summary scheme showing the pathway (calpain / P35/P25 / Cdk5) they described in their study and how roscovitine is acting on this pathway.

Answer: We have added a schematic summary of the pathways in Figure 7 which may explain how roscovitine might function (line 271-277).

Reviewer 2 Report

Hirokawa and coauthors have investigated the role of cyclin-dependent kinase-5 (CDK) driving the retinal ganglionic cell (RGS) injury after optic nerve crush (ONC). They found that Cdk5 induced phosphorylation of tau and contributed the RGS death after ONC. Administration  of Cdk5 inhibitor, Roscovitine, decreased tau-phosphorylation and prevent RGS death. The results revealed the new target, tau phosphorylation, as promising strategy for the treatment of optical nerve injury.

This is a good study on an important topic.  In general, the manuscript well prepared and the experiments well designed and mapped out.  Data are convincing and well-presented and discussed. Some of the concerns are outlined below:

  1. It is not clear rational for the dose and time of the Roscovitine administration. Time and concertation -dependent curve is very important to establish for Roscovitine.
  2. Does Roscovitine prevent the cells death? The quantification of cells death is important to include.
  3. Quantitative immunofluorescence should be performed for analyzing the upregulation of p35/p25 positive granules in    Figure 4, 5 and 6. Based on the presented images it is very difficult to conclude is it any increase in florescence described  in results.
  4. Figure 6. Are reduced number of Tuj-1 + cells associate with tau-phosphorylation? Please clarify.

Author Response

Responses to the Reviewer ï¼’.

We have revised the manuscript according to your comments and suggestions. We thank you for your constructive comments. Our responses are detailed below.

  1. It is not clear rational for the dose and time of the Roscovitine administration. Time and concertation-dependent curve is very important to establish for Roscovitine. Does Roscovitine prevent the cells death? The quantification of cells death is important to include.

Answer: Thank you. We have added some information of the IC50 for roscovitine to inhibit Cdk5/P35 in the Discussion section. Roscovitine may affect other kinases including erk1/2, but 10 µM does not reach the IC50 level. Most of the other kinases are not sensitive to roscovitine. The effects of roscovitine given after neuronal insults would be more important from a therapeutic point of view. However, the Cdk5 activity is already increased at 3 hours after induction of brain ischemia. Thus, we injected roscovitine just after the ONC (line 206-216). The neuroprotective effect of roscovitine was determined by counting the number of RGCs on day 7, and the results are shown in Figure 6B.

  1. Quantitative immunofluorescence should be performed for analyzing the upregulation of p35/p25 positive granules in Figure 4, 5 and 6. Based on the presented images it is very difficult to conclude is it any increase in florescence described in results.

Answer: We measured the fluorescein intensities of P35/P25 and tau and the results have been added in the Results section (line 127-129, 143-145, 163-167).

  1. Figure 6. Are reduced number of Tuj-1 + cells associate with tau-phosphorylation? Please clarify.

Answer: The antibody to tau for the IHC is shown to binds to total tau in Figure 6. However, the level of phosphorylated tau increased more than that of total tau from the results obtained by immunoblotting. Thus, phosphorylated tau may contribute more to the increase of the fluorescein intensities. We have added a description of this in the Results section (line 167-170).

Round 2

Reviewer 1 Report

The authors have answered to all my concerns.

Reviewer 2 Report

no comments